# Foliar Spraying of Tomato Plants with Systemic Insecticides: Effects on Feeding Behavior, Mortality and Oviposition of *Bemisia tabaci* (Hemiptera: Aleyrodidae) and Inoculation Efficiency of Tomato Chlorosis Virus

**DOI:** 10.3390/insects11090559

**Published:** 2020-08-22

**Authors:** Nathalie Kristine Prado Maluta, João Roberto Spotti Lopes, Elvira Fiallo-Olivé, Jesús Navas-Castillo, André Luiz Lourenção

**Affiliations:** 1Agronomic Institute (IAC), Centro de Fitossanidade, 13020-902 Campinas, SP, Brazil; andre.lorenzon@usp.br; 2Department of Entomology and Acarology, ESALQ, University of São Paulo, 13418-900 Piracicaba, SP, Brazil; jrslopes@usp.br; 3Instituto de Hortofruticultura Subtropical y Mediterránea ‘La Mayora’, Universidad de Málaga - Consejo Superior de Investigaciones Científicas (IHSM-UMA-CSIC), 29750 Algarrobo-Costa, Málaga, Spain; efiallo@eelm.csic.es (E.F.-O.); jnavas@eelm.csic.es (J.N.-C.)

**Keywords:** electrical penetration graph, crinivirus, probing behavior, whitefly, flupyradifurone, cyantraniliprole, acetamiprid, chemical control

## Abstract

**Simple Summary:**

The whitefly *Bemisia tabaci* (Hemiptera: Aleyrodidae) causes serious losses to vegetable, ornamental and fiber crops, including tomato plants, mainly as a vector of economically important viruses. Among the most important viruses affecting tomato is the tomato chlorosis virus (ToCV) (*Closteroviridae: Crinivirus*), which is semi-persistently transmitted by whiteflies. Effective management of this pest is crucial to reduce the spread of vector-borne diseases and to reduce crop damage and losses. In this study we evaluated the effect of systemic insecticides (cyantraniliprole, acetamiprid and flupyradifurone) on the feeding behavior, mortality and oviposition of *B. tabaci* MEAM1 and their ability to interfere with the inoculation of ToCV in tomato plants. Our findings indicate that systemic insecticides cause high mortality when compared to untreated plants. Also, we found that flupyradifurone affects stylet activities of *B. tabaci* and significantly reduce phloem ingestion, a behavior that is closely linked to the transmission of ToCV. Overall, our findings indicate that flupyradifurone may contribute to management of this pest and ToCV in tomato crops.

**Abstract:**

Tomato chlorosis virus (ToCV) is a phloem-limited crinivirus transmitted by whiteflies and seriously affects tomato crops worldwide. As with most vector-borne viral diseases, no cure is available, and the virus is managed primarily by the control of the vector. This study determined the effects of the foliar spraying with the insecticides, acetamiprid, flupyradifurone and cyantraniliprole, on the feeding behavior, mortality, oviposition and transmission efficiency of ToCV by *B. tabaci* MEAM1 in tomato plants. To evaluate mortality, oviposition and ToCV transmission in greenhouse conditions, viruliferous whiteflies were released on insecticide-treated plants at different time points (3, 24 and 72 h; 7 and 14 days) after spraying. Insect mortality was higher on plants treated with insecticides; however, only cyantraniliprole and flupyradifurone differed from them in all time points. The electrical penetration graph (DC-EPG) technique was used to monitor stylet activities of viruliferous *B. tabaci* in tomato plants 72 h after insecticide application. Only flupyradifurone affected the stylet activities of *B. tabaci*, reducing the number and duration of intracellular punctures (pd) and ingestion of phloem sap (E2), a behavior that possibly resulted in the lower percentage of ToCV transmission in this treatment (0–60%) in relation to the control treatment (60–90%) over the periods evaluated. Our results indicate that flupyradifurone may contribute to management of this pest and ToCV in tomato crops.

## 1. Introduction

The effective management of agricultural pests is highly important to minimize the damage and consequent economic losses to crops. The management of insect vectors of phytopathogens, both viruses and bacteria, is a serious challenge, due to the complex interactions among host plants, pathogens and insect vectors in their different environments [1]. These interactions become more complex in the pathosystem, involving the vector whitefly *Bemisia tabaci* (Genn.) (Hemiptera: Aleyrodidae), which is in fact a complex of cryptic species that colonize more than 600 species of plants [2,3,4], including vegetable, root and fiber crops in the families Cucurbitaceae, Euphorbiaceae, Malvaceae and Solanaceae, among others [5,6]. 

In addition to the direct damage caused by sucking sap and inducing physiological disorders, *B. tabaci* transmits different plant viruses to crops such as cassava, cotton, beans, potato and tomato [7,8,9]. Tomato chlorosis virus (ToCV) (genus *Crinivirus*, family *Closteroviridae*) is a paradigmatic example of a whitefly-transmitted emerging plant virus, infecting a large number of plants worldwide [10,11,12]. This virus is a serious problem for the cultivation of members of Solanaceae—mainly tomato—where it was first identified in 1996 in Florida, USA [13]. ToCV is transmitted semi-persistently by different species of the *B. tabaci* complex, including New World (NW, formerly biotype A), Middle East Asia Minor 1 (MEAM1, formerly biotype B) [13,14] and Meditteranean (MED, formerly biotype Q) [15,16]; as well as whiteflies of the genus *Trialeurodes, T. abutiloneus* Haldeman and *T. vaporariorum* Westwood [13]. 

Chemical control plays an important role in the management of insect-vector populations [17], reducing the number of individuals that can transmit plant pathogens in the field. However, successful chemical control requires extensive knowledge of the effects of insecticides, not only on the mortality and oviposition of the vector insect, but also on the feeding behavior associated with the acquisition and inoculation of viruses and other pathogens. In many cases, insecticides have only a limited effect on the control of viruses in spite of causing high insect mortality in treated plants. This occurs when the insecticides do not affect the feeding behavior quickly enough to prevent the insect’s stylet activities that can transmit the virus. In the case of ToCV, which is a phloem-limited virus [13,18], whiteflies must pierce the phloem cells with their stylets for transmission to occur [18]. 

The direct-current electrical penetration graph (DC-EPG) technique [19,20] is an important tool to study the feeding behavior of sucking insects, making it possible to record in real time any changes in the insect’s stylet activities caused by any environmental modification, including the application of insecticides, and thus to assess the impact on pathogen transmission [21,22]. 

In this study, we evaluated the effects of the systemic insecticides acetamiprid, cyantraniliprole and flupyradifurone on mortality, oviposition and transmission of ToCV by *B. tabaci* MEAM1 in tomato plants after different time points. We also investigated the effects of these insecticides on the feeding behavior of viruliferous whiteflies 72 h after a leaf spraying, associating changes in stylet activities with the transmission of ToCV. The results of this study increase our understanding of the effects of insecticides on the feeding behavior of whiteflies in order to determine the best chemical-control strategies to reduce their populations and the spread of the viruses they transmit.

## 2. Materials and Methods 

### 2.1. Whitefly Colony and Test Plants

A virus-free colony of *B. tabaci* MEAM1 was maintained under greenhouse conditions (temperature: 25 ± 10 °C) in insect-proof cages on cabbage (*Brassica oleracea* L. var. *acephala* DC. cv. Manteiga). Whiteflies were synchronized prior to experiments to ensure age homogeneity. Genetic identity was confirmed periodically by amplifying and sequencing the cytochrome oxidase I mitochondrial gene according to the protocol of De Barro et al. [4]. 

The ToCV isolate was collected in the municipality of Sumaré, São Paulo state, Brazil, from tomato plants (*Solanum lycopersicum* L.). The source plants of ToCV were obtained by placing non-viruliferous adult whiteflies from the colony maintained on cabbage in a 50-mL Falcon tube with a ToCV-symptomatic tomato leaf for a 48-h acquisition access period (AAP), which were then confined in groups of 30–35 insects on leaves of healthy (2–3 true-leaf stage) tomato plants (*S. lycopersicum* cv. Santa Cruz ‘Kada’), by using clip-cages, for a 5-day inoculation access period (IAP). After the IAP, leaves infested with insects were removed from the plants to eliminate eggs and nymphs, and the source plants were placed inside cages covered with voile and maintained in an insect-free greenhouse. The identity of the virus isolate was confirmed by reverse-transcription polymerase chain reaction, according to Dovas et al. [23]. 

The heathy tomato plants used in this study were grown from seeds in plastic pots containing a soil mix composed of shredded pine bark, peat, and expanded vermiculite at 60% w/w moisture content (Tropstrato HT; Vida Verde, Mogi Mirim, SP, Brazil). The plants were kept in insect-free screenhouses, where they were fertilized once a week with "Planta 100", 4 ml/L (Nutrijardim; São Carlos, SP, Brazil). 

To obtain the viruliferous insects used in the greenhouse and EPG experiments, symptomatic leaves excised from ToCV source tomato plants were placed in 50-mL Falcon tubes, with wet cotton on the tip of the petiole in order to maintain leaf turgor. Then, 40–50 healthy adult whiteflies (1–7 days old) from the colony, maintained on cabbage, were placed in each tube with an infected leaf, where they were kept for an AAP of 48 h. The insects were used in the EPG and transmission assay immediately after the AAP.

### 2.2. Stylet Activity of Bemisia tabaci MEAM1 in Plants Treated with Insecticides

The EPG technique was used to evaluate the stylet behavior of viruliferous *B. tabaci* MEAM1 in tomato plants treated with different systemic insecticides. Test plants used were treated (foliar spray) 72 h before the EPG experiment began with the selected systemic insecticides: cyantraniliprole (Benevia^®^; FMC, Campinas, SP, Brazil) at 1 mL/L; flupyradifurone (Sivanto^®^; Bayer, Paulínia, SP, Brazil) at 2 mL/L; acetamiprid (Mospilan^®^; Ihara, Sorocaba, SP, Brazil) at 0.2 g/L; or distilled water only (untreated plants). The plants were sprayed until the point of drainage using a compression sprayer (Vonder 1.5 L, Curitiba, Paraná, Brazil).

Adult females were anesthetized by confining them in glass tubes that were chilled in an ice bath for 3–5 min after being placed on a Petri dish lid that was set atop a dish filled with crushed ice under a dissecting microscope. A thin gold wire (2 cm long, 12.5 μm in diameter; EPG Systems, Wageningen, The Netherlands) was attached to the female *B. tabaci* pronotum with a droplet of silver conductive paint glue (Colloidal Silver Liquid; Ted Pella). The opposite end of the gold wire was glued to a thin copper wire (2 cm length), which was connected to the EPG probe. Another copper electrode (10 cm long, 2 mm in diameter) was inserted into the soil of the plant pot. Each whitefly was placed individually on the abaxial surface of the first tomato leaf fully expanded from top to bottom after a 1-h starvation period.

The EPG waveforms were recorded and observed on a computer screen in real-time, by using a Direct Current eight-channel EPG device, model Giga−8 d, with Stylet+ for Windows software (EPG Systems, The Netherlands). EPG signals were recorded for 10 h inside a Faraday cage (for electrical noise isolation) in a climate-controlled room (25 ± 1 °C). A total of 15 replicates per treatment were performed for ToCV-viruliferous insects in tomato plants 72 h after insecticide application.

### 2.3. Effect of Insecticides on Mortality, Oviposition and Transmission of ToCV by Bemisia tabaci MEAM1 in Tomato Plants

To determine the effects of different insecticides on ToCV transmission, healthy tomato plants (*S. lycopersicum* L. cv. Santa Cruz Kada) with 2–3 true leaves were used. The leaves were sprayed with one of the following: cyantraniliprole (Benevia^®^; FMC, Campinas, SP, Brazil) at 1 mL/L; flupyradifurone (Sivanto^®^; Bayer, Paulínia, SP, Brazil) at 2 mL/L; acetamiprid (Mospilan^®^; Ihara, Sorocaba, SP, Brazil) at 0.2 g/L; distilled water only (untreated plants). The plants were sprayed until the point of drainage using a compression sprayer (Vonder 1.5 L, Curitiba, Paraná, Brazil)

To determine the effect of the insecticides on mortality, oviposition and transmission of ToCV, 15 ToCV-viruliferous adults’ whiteflies (48 h of AAP in ToCV source plants, 1–10 days old) were released per plant, 3, 24, and 72 h, and 7 and 14 days after the insecticides were applied. Ten plants sprayed with each insecticide or with water were used per time point.

To proceed with the release of the whiteflies, each plant was individually protected with a metal cage, covered with voile to prevent the insects from escaping. Dead insects and eggs deposited on all leaves of the plant were counted 24 h after the insects were released, and live insects were removed with the aid of a manual entomological aspirator. The plants were then sprayed with deltamethrin to eliminate any remaining insects and kept uncaged in an insect-free greenhouse for a period of 4 weeks, when samples were collected to assess the ToCV infection.

The presence of ToCV RNA in tomato leaf samples was determined by tissue blot molecular hybridization. For this, freshly cross-sectioned leaf petioles were blotted on positively charged nylon membranes (Roche Diagnostics, Mannheim, Germany), and hybridized with a digoxigenin-labeled negative sense RNA probe specific for the coat protein gene, as described previously [24]. Hybridization signals were detected on X-ray film X-Omat AR (Kodak, Rochester, NY, USA) after treatment with CDP-Star (Roche Diagnostic, Mannheim, Germany) and developed following a conventional photographic process. 

### 2.4. Data Analysis

The EPG data were analyzed according to the waveforms described for whiteflies [25,26,27]: non-probing (np); intercellular apoplastic stylet pathway (C); intracellular punctures during stylet pathway phase (pd); salivation into phloem sieve elements (E1); passive phloem-sap uptake from sieve elements (E2); active intake of xylem sap (G). 

The output of EPG recordings given by the workbook of Sarria et al. [28] for each viruliferous whitefly were used to calculate the treatment mean for each EPG sequential and non-sequential variable. Variables (mean ± SE) were calculated and compared between treatments, as previously described by Backus et al. [29]. The selected variables are as follows: number of probes per insect (NPI); number of waveform events per insect (NWEI); total waveform duration (min) per insect (WDI); waveform duration (min) per event (WDE). The sequential variables analyzed were: "Time to 1st probe from start of EPG", "No. of probes to the 1st E1", "No. of probes after 1st E", "Time from start of EPG to 1st E" and "Time from 1st probe to 1st E".

Before analysis, the normality and homogeneity of variance were checked. The data were transformed when necessary with ln (x + 1) or √ (x + 1) to reduce heteroscedasticity and improve normal distribution. All parameters were analyzed with a parametric Tukey’s test (*p* < 0.05) (data with normal distribution) or a nonparametric Kruskal–Wallis, H test (*p* < 0.05) (data did not follow a normal distribution according to the Shapiro–Wilk normality test). The mortality data (time points 3 h and 24 h) and oviposition data (time points 24 h, 7 and 14 days) were transformed (ln(x + 1)) and were analyzed by Tukey’s test. The mortality data (time points 72 h, 7 and 14 days) and oviposition data (time points 3 and 72 h) did not follow a normal distribution even after transformation and were analyzed using the Kruskall–Wallis test. A chi-squared test was used to analyze the ToCV transmission rate. All data were analyzed using IBM SPSS Statistics, version 22.0 software [30]. 

## 3. Results

### 3.1. Stylet Activity of B. tabaci on Plants Treated with Insecticides

In the EPG experiment, changes in the probing behavior of viruliferous whiteflies were observed in plants treated with flupyradifurone, since they performed fewer intercellular stylet pathways (NWEI; waveform C) (H = 8.98; df = 3; *p* = 0.03), non-probing (NWEI; waveform np) (H = 7.83; df = 3; *p* = 0.04) and probes (NPI) (H = 7.88; df = 3; *p* = 0.04) with their stylets, compared to the control treatment and also to the other insecticide treatments (Figure 1a). However, there was no difference between treatments regarding the mean total duration (WDI) of events C (H = 3.22; df = 3; *p* = 0.36), np (H = 4.35; df = 3; *p* = 0.23) and probes (H = 4.35; df = 3; *p* = 0.23) (Figure 1b).

Regarding the parameters related to phloem activities, which are directly associated with the inoculation of ToCV, in the plants treated with flupyradifurone, whiteflies performed fewer intracellular punctures (waveform pd) than in the control treatment (H = 8.93; df = 3; *p* = 0.03) (Figure 1c). However, the number of times that the insects salivated in the phloem vessels (waveform E1) (H = 3.77; df = 3; *p* = 0.29) and ingested phloem sap (waveform E2) (H = 2.19; df = 3; *p* = 0.53) did not differ between the treated and control plants (Figure 1c).

The mean total duration (WDI) of waveform pd was shorter in plants treated with flupyradifurone (H = 7.89; df = 3; *p* = 0.04), as was the total duration of phloem sap ingestion (waveform E2) (H = 7.99; df = 3; *p* = 0.04) (Figure 1d).

In the plants treated with acetamiprid and cyantraniliprole, there were no differences in the non-phloem and phloem parameters of the number (NWEI) and duration (WDI) of events per insect, compared to the control treatment (Figure 1).

Regarding the duration of the waveform per event (WDE), differences were observed in the durations of waveform C (H = 14.27; df = 3; *p* < 0.01), np (H = 64.14; df = 3; *p* < 0.01) and probes (H = 12.47; df = 3; *p* < 0.01); the duration of these events was longer in plants treated with flupyradifurone (Figure 2a). This indicated that, although the number (NWEI) of times that the individual whiteflies performed these events was significantly lower in this treatment (Figure 1a), the duration of each event (WDE) was longer (Figure 2a). Regarding the phloem parameters of WDE, no differences were observed between treatments: (WDE pd: H = 4.62; df = 3; *p* = 0.20); (WDE E1: F = 0.12; df = 3; *p* = 0.89); (WDE E2: F = 0.51; df = 3; *p* = 0.62). Regarding the WDE of E1 and E2, in the flupyradifurone treatment there was only one event of each type of waveform, and because it was not possible to generate a mean and error bar, the events were not plotted on the graph (Figure 2b).

No significant differences were observed for any sequential variable analyzed (Table 1).

### 3.2. Effect of Insecticides on Mortality, Oviposition and ToCV Transmission by Bemisia tabaci MEAM1 in Tomato Plants

The insecticides applied to tomato plants by foliar spraying proved to affect ToCV transmission by *B. tabaci* MEAM1, as well as causing insect mortality and reduction oviposition. The effects varied according to the insecticide and the time after application (Figure 3, Figure 4 and Figure 5).

In the time points 3 h (F = 4.85; df = 3; *p* < 0.01), 24 h (F = 5.45; df = 3; *p* = 0.03) 7 days (H = 13.04; df = 3; *p* < 0.01) and 14 days (H = 9.05; df = 3; *p* = 0.03) after the spraying, higher mortality was observed in plants treated with flupyradifurone and cyantraniliprole when compared to the control treatment. Higher mortality in all treatments with insecticides was observed only 72 h after spraying (H = 18.20; df = 3; *p* < 0.01) (Figure 3). The number of live insects found per plant varied in the evaluated times: 12.4–14.1 insects in the control; 8.1–12.6 in plants treated with cyantraniliprole; 9.0–1.3.0 in the treatment of acetamiprid and 8.5–12.1 in plants treated with flupyradifurone. 

A lower percentage of ToCV transmission was observed only in plants sprayed with flupyradifurone, up to 7 days after spraying (Figure 4). In the time points 3 h (χ2 = 9.04; df = 3; *p* = 0.03), 24 h (χ2 = 20.07; df = 3; *p* < 0.01), 72 h (χ2 = 18.32; df = 3; *p* < 0.01) and 7 days (χ2 = 19.67; df = 3; *p* < 0.01) after the application of insecticides, the percentages of infected plants ranged from 0 to 30% in the treatment with flupyradifurone, contrasting with 60–90% in the control treatment (Figure 4).

For plants where insects were released 14 days after the application of insecticides, no differences were observed regarding the percentage of infected plants, since the transmission rate was higher than 60% in all treatments (χ2 = 3.972; df = 3; *p* = 0.26) (Figure 4).

The oviposition of *B. tabaci* MEAM1 was also affected by insecticides; 3 h after spraying, whiteflies deposited significantly fewer eggs on the plants treated with acetamiprid and flupyradifurone (H = 11.99; df = 3; *p* < 0.01). Twenty-four hours after spraying, fewer eggs were observed in all treatments compared to the control (F = 3.34; df = 3; *p* = 0.03). At 7 days after spraying, however, only the plants treated with acetamiprid differed from the control (F = 3.43; df = 3; *p* = 0.03) (Figure 5).

In the time point of 72 h (H = 0.57; df = 3; *p* = 0.91) and 14 days (F = 0.14; df = 3; *p* = 0.94) after the spraying of insecticides, no differences were observed in the number of eggs deposited on the plants treated in relation to the plants of the control treatment (Figure 5).

## 4. Discussion

Many plant viruses depend on vectors to spread to new crop areas, and therefore the success of transmission and the consequent colonization of new host plants is closely associated with the probing behavior of their vectors [18,31,32,33]. ToCV, as the rest of criniviruses, is restricted to phloem vessels [13,18], a location where whiteflies feed. The use of systemic insecticides is one of the most widely available management measures for controlling populations of *B. tabaci* in an attempt to reduce the spread of the viruses they transmit, including ToCV, since these insecticides can reduce the transmission and spread of vector-transmitted pathogens to various crops [22,34,35]. 

We found a clear difference between the insecticides tested in their ability to affect the probing behavior, transmission of ToCV, oviposition and mortality of *B. tabaci* MEAM1. In general, the insecticides, mainly cyantraniliprole and flupyradifurone, caused higher mortality than the control treatment. However, the cyantraniliprole treatment did not decrease the rate of ToCV transmission, since the percentage of infected plants did not differ from the control treatment in any period evaluated. In insecticide-treated plants, oviposition was lower up to 24 h after application, with a trend toward no effect for periods equal to or longer than 72 h. The only exception was the acetamiprid treatment, in which an effect on oviposition was observed when the insects were released 7 days after spraying.

Gouvea et al. [36] found that foliar applications and drenching with cyantraniliprole did not significantly reduce the primary transmission of the begomovirus (family *Geminiviridae*) tomato severe rugose virus to tomato plants by *B. tabaci* MEAM1 but were able to reduce secondary transmission. However, these authors observed higher adult mortality of whiteflies in the plants treated with cyantraniliprole (50–54%) compared to plants treated with only water (10.4%) 48 h after the insects were confined on these plants, results close to those of the present study.

Our results also showed that the stylet activities of *B. tabaci* MEAM1 are not always altered in plants treated with insecticides, but when there is an effect on probing behavior, the rate of transmission is reduced. It may be that certain insecticides do not alter the stylet activities associated with the transmission of pathogens, and moreover are not able to kill the insects fast enough to prevent them from feeding and consequently inoculating ToCV into new plants.

In our study, we observed that, in plants treated with flupyradifurone, the whiteflies ingested phloem sap for shorter periods (waveform E2) and the duration of intracellular punctures (pd) was also shorter than in the control treatment. These behaviors, together with other stylet activities, possibly resulted in a lower rate of ToCV inoculation into these plants. A remarkable effect on probing and phloem activities of *B. tabaci* MED was observed by Garzo et al. [37] in tomato plants treated with flupyradifurone and cyantraniliprole. In addition, the authors observed a reduction in the transmission of tomato yellow leaf curl virus (TYLCV) (family *Geminiviridae*, genus *Begomovirus*) to treated tomato plants. 

Carmo-Sousa et al. [22], studying *Diaphoria citri* Kuwayama (Hemiptera: Liviidae), and Butler et al. [38], studying *Bactericera cockerelli* (Sulc) (Hemiptera: Triozidae), also found a pronounced effect of systemic insecticides on the duration of phloem sap ingestion (waveform E2) and reductions in the inoculation of phytopathogenic bacteria. 

Castle et al. [35] found that no cantaloupe melon plants were infected with another crinivirus, cucurbit yellow stunting disorder virus, when sprayed with flupyradifurone (Sivanto), for any of three *B. tabaci* MEAM1 densities tested (3, 10 and 30 insects), and observed only one infected plant in the acetamiprid treatment (Assail 70WP), at the highest insect density. In our study, we found no effects of acetamiprid on the probing behavior and the rate of ToCV transmission to tomato plants. We observed only that 72 h after the acetamiprid spraying, insect mortality was higher in these plants and oviposition was lower (at 3 h, 24 h and 7 days after the application) compared to the control treatment.

Roditakis et al. [39] found a reduction of 85% in the transmission of tomato yellow leaf curl virus by *B. tabaci* MED in plants treated with flupyradifurone, compared to 25% in plants treated with thiamethoxam and 100% in the control treatment. Promising results were also obtained by Smith and Giurcanu [40], who observed a drastic reduction in the percentage of tomato plants expressing symptoms of the TYLCV begomovirus when treated with flupyradifurone (0–5%), compared to the control (75–100%) in periods 3, 7 and 14 days after the application of the insecticide.

We found no effect on the stylet activities of *B. tabaci* MEAM1 in plants treated with cyantraniliprole, nor any reduction in the rate of ToCV transmission, although whitefly mortality was higher compared to the control. However, Civolani et al. [41] observed that, in plants treated with cyantraniliprole, non-viruliferous *B. tabaci* Q2 (a variant of the MED species) were not able to reach the phloem vessels in tomato plants.

This could indicate that the presence of the virus in the insect’s body/mouthparts alters the behaviors as strategies of the pathogens to promote their own dissemination to new host plants, as has been observed in other studies [31,42,43,44]. 

As previously reported, pathogens have the ability to alter the behavior and performance of their vectors [31,45,46], which can behave differently if they are viruliferous or not or encounter infected plants. In addition, the effects of insecticides on the cryptic species of *B. tabaci* may vary widely, as these species differ in many characteristics, such as the species of endosymbionts, virus transmission efficiency, adaptation to hosts, resistance to insecticides, and responses to pathogens [46,47,48,49]. 

The results of the present study demonstrate that foliar applications of systemic insecticides are useful to protect tomato crops against feeding by *B. tabaci* and to reduce the transmission of ToCV to new host plants. 

## 5. Conclusions

The obtained results indicated that systemic insecticides play an important role in reducing whitefly populations and tomato chlorosis virus inoculation by *Bemisia tabaci* MEAM1. Only the insecticide flupyradifurone may potentially reduce the inoculation of ToCV into tomato plants, both by increasing insect mortality and by altering stylet activities associated with the transmission of this crinivirus, and can be an important tool in the management of this insect vector in tomato crops. 

## Figures and Tables

**Figure 1 insects-11-00559-f001:**
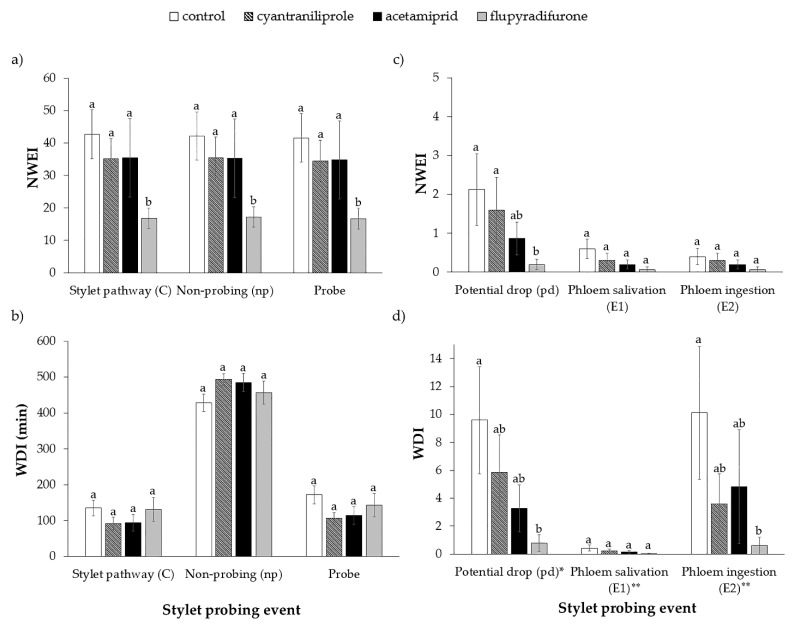
Number of waveform events per inset (NWEI) and total waveform duration per insect (WDI) (mean ± SE) of non-phloem (**a**,**b**) and phloem (**c**,**d**) EPG variables of ToCV-viruliferous *Bemisia tabaci* MEAM1 in tomato plants 72 h after insecticide application. Statistical comparisons between treatments for each parameter were made by non-parametric Kruskal–Wallis test. Means that share the same letter for each parameter are not significantly different. (*p* > 0.05). * Total duration of potential drop (pd) expressed in seconds; ** Total duration of E1 and E2 expressed in minutes, but it was used the same scale in the graphic.

**Figure 2 insects-11-00559-f002:**
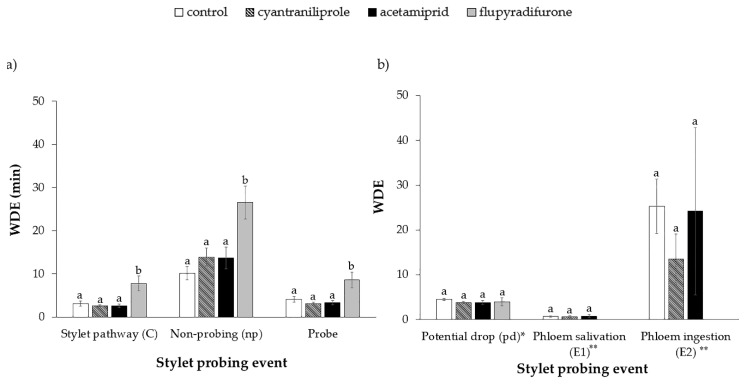
Waveform duration per event (WDE) (mean ± SE) of non-phloem (**a**) and phloem (**b**) EPG events of ToCV-viruliferous *Bemisia tabaci* MEAM1 in tomato plants 72 h after insecticide application. Statistical comparisons between treatments for each parameter were made by non-parametric Kruskal–Wallis test or parametric Tukey’s test. Means that share the same letter for each parameter are not significantly different. (*p* > 0.05). * Total duration of potential drop (pd) expressed in seconds; ** Total duration of E1 and E2 expressed in minutes, but it was used in the same scale in the graphic. Flupyradifurone data for WDE of E1 and E2—we have only one set of data and its standard error is not available.

**Figure 3 insects-11-00559-f003:**
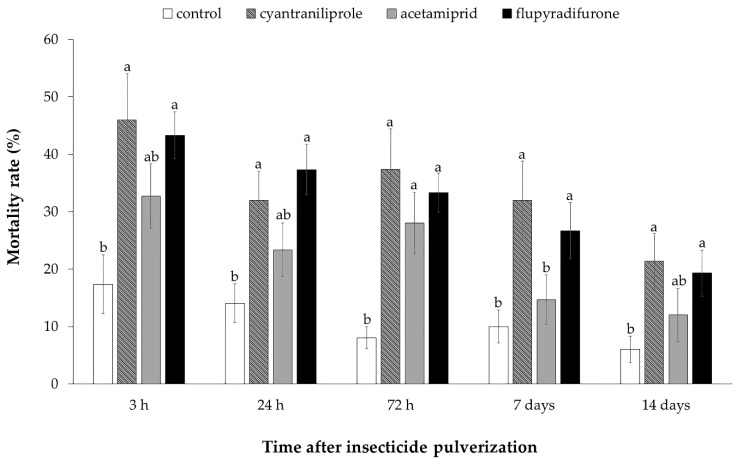
Mortality rate (%) of ToCV-viruliferous *Bemisia tabaci* MEAM1 in different time points after insecticide pulverization (AIP). Means that share the same letter are not significantly different. (*p* > 0.05) by non-parametric Kruskal–Wallis test (for non-Gaussian distribution) or Tukey’s test (for Gaussian distribution).

**Figure 4 insects-11-00559-f004:**
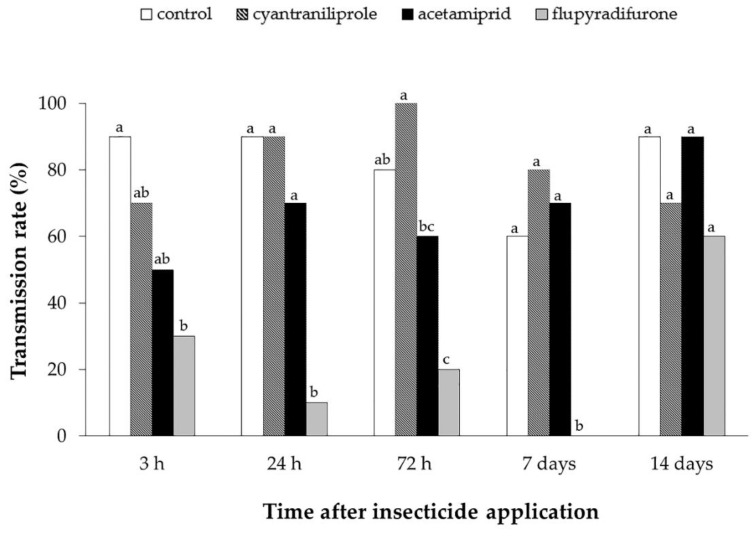
Transmission rate (%) of tomato chlorosis virus by *Bemisia tabaci* MEAM1 in different periods after insecticide pulverization (AIP). Means that share the same letter in the same time point are not significantly different. (*p* > 0.05) according to the chi-square (X2) test.

**Figure 5 insects-11-00559-f005:**
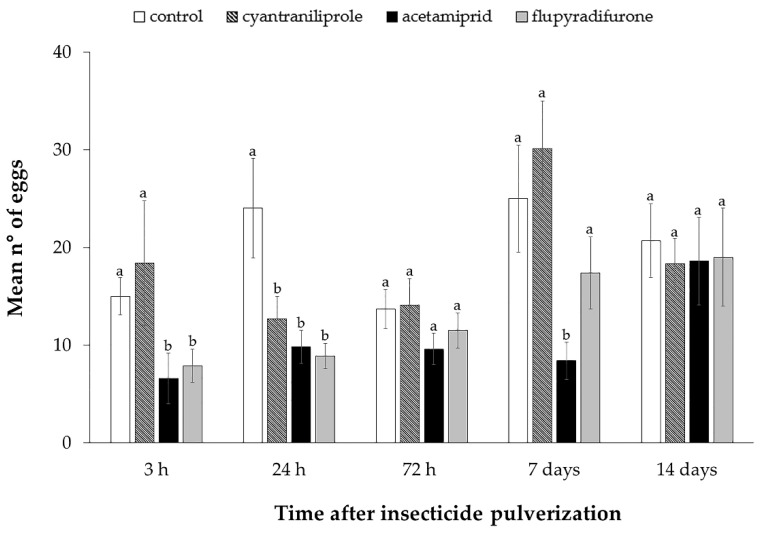
Mean number of eggs laid ± SE by ToCV-viruliferous *Bemisia tabaci* MEAM1 on tomato plants at different time points (3, 24, 72 h, 7 and 14 days) after insecticide sprays. Means that share the same letter are not significantly different. *p* > 0.05 by non-parametric Kruskal–Wallis test (for non-Gaussian distribution) or Tukey’s test (for Gaussian distribution).

**Table 1 insects-11-00559-t001:** Mean (± SEM) of sequential EPG variables for 10-h recordings of the probing and feeding behavior of ToCV-viruliferous *Bemisia tabaci* MEAM1 on tomato plants that were sprayed with different insecticide treatments 72 h earlier (time expressed in minutes).

Sequential EPG Variables ^a^	Control n = 15	Cyantraniliprole n = 15	Acetamiprid n = 15	Flupyradifurone n = 15	H	df	*p*
Time to 1st probe from start of EPG	164.19 ± 45.82 a	199.45 ± 35.64 a	200.33 ± 58.93 a	125.64 ± 38.96 a	2.74	3	0.43
N° of probes to the 1st E1	7.0 ± 3.24 a	1.80 ± 1.08 a	10.53 ± 6.40 a	0.27 ± 0.27 a	3.77	3	0.29
N° of probes after first E	7.20 ± 3.44 a	2.20 ± 2.13 a	1.27 ± 1.27 a	0.20 ± 0.20 a	3.81	3	0.28
Time from start of EPG to 1st E	487.91 ± 48.58 a	538.24 ± 37.43 a	574.91 ± 17.55 a	564.83 ± 35.17 a	3.24	3	0.36
Time from 1st probe to 1st E	323.72 ± 48.39 a	338.78 ± 46.86 a	374.58 ± 59.97 a	439.18 ± 47.91 a	3.96	3	0.26

^a^ Statistical comparison between treatments for each variable were made by non-parametric Kruskal–Wallis test (data with non-Gaussian distribution). Means that share the same letter for each sequential variable are not significantly different. (*p* > 0.05).

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
