# Peer review of "Foliar Spraying of Tomato Plants with Systemic Insecticides: Effects on Feeding Behavior, Mortality and Oviposition of Bemisia tabaci (Hemiptera: Aleyrodidae) and Inoculation Efficiency of Tomato Chlorosis Virus"

_insects, 2020, doi:10.3390/insects11090559_

Round 1

Reviewer 1 Report

This paper assess the effect of the application of chemical insecticides with a systemic activity on the transmission of a non-persistently transmitted virus vectored by B. tabaci. The MEAM1 species was used for experiments, which is one of the most economically relevant ones. The paper is generally well organized and clearly describes the methods applied and the results that were obtained. The discussion nicely comments the outcome of this research, identifying the best active substance that could be used for significantly limit the spread of ToCV by the vector. Even though chemical control is not a very innovative approach, also in consideration of the need to reduce the environmental impact of agriculture, yet it still often the main available control option, therefore the contribution to knowledge on this topic certainly deserves attention. I only have few minor suggestions to improve the paper:

1) In the materials and methods, some steps are repeatedly described due to the manuscript organization. For example, L 11-113 repeats the concepts expressed at L98-103 and could be eliminated. Moreover, the foliar treatment with the 3 active substances is repeated twice in paragraph 2.2 and 2.3 (L105-110 and L130-135). In this case, I suggest adding a separate short paragraph on plant treatment before the stylet activity and mortality oviposition and transmission ones.

2) I remark that Fig. 3 and 5 are missing, and so it was quite difficult to me to appreciate the actual role of mortality / oviposition VS alteration of feeding behaviour. Moreover, when presenting mortality rates I suggest to apply the Abbot correction, to most properly compare the treatments.

3) I believe that a major aim of this work is to clearly demonstrate that affecting the insect feeding behaviour is a key factor for the control of indirect (virus-related) damage due to B. tabaci, which is often the main one. In the discussion, the authors briefly state that only one of the products significantly reduces transmission even though all tested substances caused high mortality, probably because only flupyradifurone alters the feeding behaviour (L294-297). This is a very important aspect in non-persistent transmission models, since the lack of latency period implies that mortality induction may not be quick enough to avoid transmission. In order to more strongly express this aspect, I suggest to add in the results a short mention to the mean number of live insects found per plant, in the paragraph where the transmission efficiency is compared among treatments. In this way, it would be possible to assess if the reduced transmission is really due only to probing reduction rather than to a lower number of live insects that could inoculate the ToCV into the plant (I suppose so, even though I did not see the raw data).

Author Response

Dear Editor: We are returning a revised paper based on the suggestions raised by the reviewers of our first revised version of the manuscript. See below specific answers (point by point):

Response to Reviewer 1 Comments

We appreciate the valuable suggestions. We made all the revisions suggested by the reviewers in the text using “track changes”.

Point 1: In the materials and methods, some steps are repeatedly described due to the manuscript organization. For example, L 11-113 repeats the concepts expressed at L98-103 and could be eliminated. Moreover, the foliar treatment with the 3 active substances is repeated twice in paragraph 2.2 and 2.3 (L105-110 and L130-135). In this case, I suggest adding a separate short paragraph on plant treatment before the stylet activity and mortality oviposition and transmission ones. 

Response 1: DONE. We removed the sentence L111-113

Point 2: I remark that Fig. 3 and 5 are missing, and so it was quite difficult to me to appreciate the actual role of mortality / oviposition VS alteration of feeding behaviour. Moreover, when presenting mortality rates, I suggest to apply the Abbot correction, to most properly compare the treatments.

Response 2: Figures 3 and 5 were inserted in the text, I believe that due to the formatting they were supported, lacking in the first version of the manuscript. We applied the Abbott´s corrections in the mortality rate data (figure attached with the pdf file), but all authors believe that the figure becomes clearer when there is a control bar, without correcting the data. But if the reviewer believes that it will be essential to put the figure using the Abbott´s correction, we can modify it without problems. Also, we reviewed the mortality data and detected a miscalculation in one of the treatments, and so we performed a new statistical analysis.

Point 3: I believe that a major aim of this work is to clearly demonstrate that affecting the insect feeding behaviour is a key factor for the control of indirect (virus-related) damage due to B. tabaci, which is often the main one. In the discussion, the authors briefly state that only one of the products significantly reduces transmission even though all tested substances caused high mortality, probably because only flupyradifurone alters the feeding behaviour (L294-297). This is a very important aspect in non-persistent transmission models, since the lack of latency period implies that mortality induction may not be quick enough to avoid transmission. In order to more strongly express this aspect, I suggest to add in the results a short mention to the mean number of live insects found per plant, in the paragraph where the transmission efficiency is compared among treatments. In this way, it would be possible to assess if the reduced transmission is really due only to probing reduction rather than to a lower number of live insects that could inoculate the ToCV into the plant (I suppose so, even though I did not see the raw data).

Response 3: We includes a short paragraph mentioning the mean number of live insects per treatment in the different time points. Mortality rates were not very high in any of the treatments, although significant differences were found at all time points, so it is believed that variations in the transmission rate are mainly due to changes in the feeding behavior and secondly due to mortality.

The number of live insects found per plant varied in the evaluated times: 12.4 – 14.1 insects in the control; 8.1 – 12.6 in plants treated with cyantraniliprole; 9.0 – 1.3.0 in the treatment of acetamiprid and 8.5 - 12.1 in plants treated with flupyradifurone).” (L236-238).

Reviewer 2 Report

The manuscript investigates different effects of the foliar spraying with the systemic insecticides on feeding behavior, mortality, oviposition, and transmission efficiency of Tomato chlorosis virus (ToCV) by whitefly MEAM1 on tomato plants. The authors tested the effects of insecticides on feeding and probing behavior of whitefly, and transmission and found flupyradifurone was one of the most effective insecticides which show the reduction in feeding activities to explain lower ToCV transmission. The authors also found some effects of acetamiprid and flupyradifurone on the mortality/oviposition of whitefly. The results form this manuscript will provide information regarding the effects of insecticides on the behavior and biology of whitefly regarding virus spread. Overall, designs and analyses of experiments seemed appropriate. The presentation of methods and results was clear, and well-described (except missing figures). 

Line 104: Where there was the same start time for EPG experiments each day? Please mention the input resistance used.

Line 106: What kind of ‘tomato plants’ should it be ‘un-infected tomato plants’? What cultivar of tomato?

Line 122: How was the whitefly starved for 1 hour, was it just above the leave? Please explain.

Line 138: What was the reason to select these times to release whitefly?

Line 141: Was it always possible to find/recover the dead whitefly on the plant?

Line 168: What was the test for mortality, and oviposition? Please make the data analysis section clearer for the reader to understand what transformation and analysis were done for mortality and oviposition.

Line 169: Where the data from 2.3 were sliced by time i.e. where the data for mortality, transmission, and oviposition were separately analyzed for each time?

Figures 1d and 2 d: If possible, I would recommend having a potential drop (pd) figure separately, as pd is seconds and other phloem phases in minutes.

Figure 4: Please put the standard error bar in the bar graphs.

Figure 5: Not available

Author Response

Dear Editor: We are returning a revised paper based on the suggestions raised by the reviewers of our first revised version of the manuscript. See below specific answers (point by point):

Response to Reviewer 2 Comments

We appreciate the valuable suggestions. We made all the revisions suggested by the reviewers in the text using “track changes”.

Line 104: Where there was the same start time for EPG experiments each day? Please mention the input resistance used.

Response: All EPG records were started between 9 and 9:30 am.

Line 106: What kind of ‘tomato plants’ should it be ‘un-infected tomato plants’? What cultivar of tomato?

Response: Tomatoes considered healthy were grown in a greenhouse free of insects of any kind, and without the occurrence of viruses. We used the cultivar Santa Cruz ‘Kada’ (L88-89).

Line 122: How was the whitefly starved for 1 hour, was it just above the leave? Please explain.

Response: We considered 1 hour of starved between the time when we collected the viruliferous whiteflies from the source plants, attached the gold wire in the whitely pronotum and placed on the plant surface to start EPG records. As this time passes, there may be an effect on the feeding behavior of these insects due to dehydration. In addition, 1 hour of starvation encourages the insect to probe/feed when we put it on the plants.

Line 138: What was the reason to select these times to release whitefly?

Response: The times selected to release the whiteflies on the treated plants were defined based on conversations with an insecticide specialist from ESALQ (University of São Paulo-Brazil), trying to contemplate periods when the insecticides would be only in the form of contact (short times), periods when the insecticides are systemic in plants vessels (24 - 72h) where more control efficiency was expected, and long times where less control efficiency is expected (7-14 days).

Line 141: Was it always possible to find/recover the dead whitefly on the plant?

Response: The dead insects were easily collected from the plants, as a black cardboard was placed on the soil surface of each plant, preventing the insects from falling into the soil. In addition, many insects were dead on the leaves of the tomato plants with the stylets inserted into the plant tissue. To avoid escape of live insects during the counting of dead and live insects, each plant was evaluated inside a black chamber with lighting.

Line 168: What was the test for mortality, and oviposition? Please make the data analysis section clearer for the reader to understand what transformation and analysis were done for mortality and oviposition.

Response: We includes a brief explanation about the statistical analysis used in the mortality data. “The mortality data (time points 3 h and 24 h) and oviposition data (time points 24 h, 7 and 14 days) were transformed (ln(x+1)) and were analyzed by Tukey´s test. The mortality data (time points 72h, 7 and 14 days) and oviposition data (time points 3 and 72 h) did not follow a normal distribution even after transformation and were analyzed using the Kruskall-Wallis test.” (L166-170- 2.4. Data analysis).

Line 169: Where the data from 2.3 were sliced by time i.e. where the data for mortality, transmission, and oviposition were separately analysed for each time?

Response: The data were analysed separately because the intention was to compare the untreated plants (control) vs. treatments with insecticides within the same time point. Furthermore, at each time point after the application of the insecticides different plants and insects were used. That is, at 3h the insects and plants are different from the 24h time, not being an accumulated count.

Figures 1d and 2 d: If possible, I would recommend having a potential drop (pd) figure separately, as pd is seconds and other phloem phases in minutes.

Response: I appreciate the suggestion, however, although the “pd” values ​​should be understood as seconds and E1 and E2 as minutes, the scale is the same in the graph. We believe that it is not worth making a figure just for the parameter “pd”, since it would take up a lot of space and is not the most important parameter for so much highlight. We explained better in the Figure footnote (Figure 1 and 2).

Figure 4: Please put the standard error bar in the bar graphs.

Response: In this case there is no error bar, because the plants were either infected or not. Similar data and figures can be seen in:

Garzo, E et al (2020) Feeding behavior and virus-transmission ability of insect vectors exposed to systemic insecticides. Plants 2020, 9, 895. https://doi.org/10.3390/plants9070895

Figure 5: Not available

Response: Figures 3 and 5 were inserted in the text, I believe that due to the formatting they were supported, lacking in the first version of the manuscript. We apologize.

Reviewer 3 Report

Vegetables are often impacted by Whitefly-transmitted viruses. Management of these viruses has largely relied on using insecticides. This manuscript by Maluta et al. assesses how a few commonly used insecticides suppress whitefly feeding/probing and how these effects can in turn reduce a crinivirus transmission. The mechanics of whitefly probing and effects on whitefly fitness have largely been addressed in this manuscript.  The information presented in this manuscript could be useful for vector and/or virus management and complement existing literature. The manuscript needs minor revisions. The revisions are directly tracked in the manuscript. The manuscript can be accepted for publication pending completion of the suggested minor revisions.

Author Response

Response to Reviewer 3 Comments

Vegetables are often impacted by Whitefly-transmitted viruses. Management of these viruses has largely relied on using insecticides. This manuscript by Maluta et al. assesses how a few commonly used insecticides suppress whitefly feeding/probing and how these effects can in turn reduce a crinivirus transmission. The mechanics of whitefly probing and effects on whitefly fitness have largely been addressed in this manuscript.  The information presented in this manuscript could be useful for vector and/or virus management and complement existing literature. The manuscript needs minor revisions. The revisions are directly tracked in the manuscript. The manuscript can be accepted for publication pending completion of the suggested minor revisions.

Response: We appreciate the valuable suggestions. We made all the minor revisions suggested by the reviewers in the text using” track changes”.
